# Nitrite Reductase Activity of Ferrous Nitrobindins: A Comparative Study

**DOI:** 10.3390/ijms24076553

**Published:** 2023-03-31

**Authors:** Giovanna De Simone, Alessandra di Masi, Grazia R. Tundo, Massimo Coletta, Paolo Ascenzi

**Affiliations:** 1Dipartimento di Scienze, Università Roma Tre, 00146 Roma, Italy; 2Dipartimento di Scienze Cliniche e Medicina Traslazionale, Università di Roma Tor Vergata, 00133 Roma, Italy; 3IRCCS Fondazione Bietti, 00198 Roma, Italy; 4Laboratorio Interdipartimentale di Microscopia Elettronica, Università Roma Tre, 00146 Roma, Italy

**Keywords:** *Mycobacterium tuberculosis* nitrobindin, *Arabidopsis thaliana* nitrobindin, *Danio rerio* nitrobindin, *Homo sapiens* nitrobindin, nitrite reductase activity, kinetics

## Abstract

Nitrobindins (Nbs) are all-β-barrel heme proteins spanning from bacteria to *Homo sapiens*. They inactivate reactive nitrogen species by sequestering NO, converting NO to HNO_2_, and promoting peroxynitrite isomerization to NO_3_^−^. Here, the nitrite reductase activity of Nb(II) from *Mycobacterium tuberculosis* (*Mt*-Nb(II)), *Arabidopsis thaliana* (*At*-Nb(II)), *Danio rerio* (*Dr*-Nb(II)), and *Homo sapiens* (*Hs*-Nb(II)) is reported. This activity is crucial for the *in vivo* production of NO, and thus for the regulation of blood pressure, being of the utmost importance for the blood supply to poorly oxygenated tissues, such as the eye retina. At pH 7.3 and 20.0 °C, the values of the second-order rate constants (i.e., *k*_on_) for the reduction of NO_2_^−^ to NO and the concomitant formation of nitrosylated *Mt*-Nb(II), *At*-Nb(II), *Dr*-Nb(II), and *Hs*-Nb(II) (Nb(II)-NO) were 7.6 M^−1^ s^−1^, 9.3 M^−1^ s^−1^, 1.4 × 10^1^ M^−1^ s^−1^, and 5.8 M^−1^ s^−1^, respectively. The values of *k*_on_ increased linearly with decreasing pH, thus indicating that the NO_2_^−^-based conversion of Nb(II) to Nb(II)-NO requires the involvement of one proton. These results represent the first evidence for the NO_2_ reductase activity of Nbs(II), strongly supporting the view that Nbs are involved in NO metabolism. Interestingly, the nitrite reductase reactivity of all-β-barrel Nbs and of all-α-helical globins (e.g., myoglobin) was very similar despite the very different three-dimensional fold; however, differences between all-α-helical globins and all-β-barrel Nbs suggest that nitrite reductase activity appears to be controlled by distal steric barriers, even though a more complex regulatory mechanism can be also envisaged.

## 1. Introduction

All-α-helical heme proteins are pivotal for O_2_, NO, and CO sensing, storing, transport, and chemistry [1,2,3,4]. Classical globins, including hemoglobin (Hb) and myoglobin (Mb), are composed of eight α-helical segments that are arranged in a 3/3 fold around the heme; the A, B, and E α-helices face the heme on one side and the F, G, and H α-helices on the other side [2,3,4,5,6].

Over the last 25 years, additional structural folds have been observed in heme proteins, such as a subset of all-α-helical globins, called truncated hemoglobins (trHbs), which were found in bacteria, plants, and in some unicellular eukaryotes [7]. TrHbs display a 2/2-fold, in which two anti-parallel pairs of helices (i.e., B/E and G/H) enclose the heme [7,8,9,10,11,12,13]. In 3/3 and 2/2 all-α-helical heme proteins, the fifth ligand of the metal center is the Nε atom of the so-called “proximal” histidyl residue; in most cases, a second histidyl side chain faces the heme–Fe-atom at the heme distal site [1,2,3,4,5,6].

More recently, all-β-barrel heme proteins (i.e., nitrophorins (NPs) and nitrobindins (Nbs)) have been discovered and characterized from both structural and functional viewpoints. Interestingly, NPs are present only in the salivary gland of *Rhodnius prolixus* whereas Nbs span from bacteria to *Homo sapiens* [13,14,15,16,17,18]. Although in NPs and Nbs the heme is anchored to the protein moiety by a proximal His residue, forming the fifth axial ligand of the Fe atom, they are predominantly in the ferric form [13,14,15,16,17,18,19]. On the other hand, the histidyl residue, which is facing the heme distal side in most all-α-helical globins, is absent in NPs and Nbs, thus preventing the heme–Fe-atom six-coordination and the consequent inactivation of the metal center [19].

Ferric and ferrous Nbs (Nb(III) and Nb(II), respectively) bind reversibly to NO with similar combination rate constants ((1.4 ± 0.4) × 10^6^ M^−1^ s^−1^ and (1.5 ± 0.6) × 10^6^ M^−1^ s^−1^, respectively); the very different values of the dissociation rate constant ((4.6 ± 2.6) × 10^1^ s^−1^ and (5.7 ± 2.5) × 10^−2^ s^−1^, respectively) are at the root of the different affinities ((3.5 ± 2.5) × 10^−5^ M and (3.5 ± 2.5) × 10^−8^ M, respectively) [19,20,21,22]. Of note, NO binding to the sixth coordination position of the heme–Fe-atom induces the cleavage or the severe weakening of the fifth proximal His-Fe(II) bond, at a neutral pH; this process occurs only at low pH or in the presence of allosteric effectors in Mb and Hb, respectively [23,24]. Moreover, at alkaline pHs, NO facilitates the reduction of the heme–Fe(III)-atom leading to the nitrosylation of the ferrous metal center (Nb(II)-NO) with a OH^−^-dependent reaction via the transient formation of ferric nitrosylated Nb (Nb(III)-NO); concomitantly, one equivalent of HNO_2_ is produced [20,22]. Furthermore, Nbs catalyze the isomerization of peroxynitrite to NO_3_^−^ and NO_2_^−^ in the absence and presence of CO_2_ [19,25,26]. Lastly, O_2_ scavenging by Nb(II)-NO leads to NO_3_^−^ and Nb(III) via the transient formation of a Nb(III)-NO(O)O adduct [24].

Here, the nitrite reductase activity of Nb(II) from *Mycobacterium tuberculosis* (*Mt*-Nb(II)), *Arabidopiss thaliana (At*-Nb(II)), *Danio rerio* (*Dr*-Nb(II)), and *Homo sapiens* (*Hs*-Nb(II)) is reported and analyzed in parallel with the kinetics of several heme proteins spanning from globins to cytochrome c [27,28,29,30].

Although the physiological function(s) of Nbs remain(s) still largely obscure, it is worth remarking that all these data indeed suggest that Nbs might be involved in NO signaling and metabolism. Indeed, nitrosative stress plays a pivotal role in the onset and progression of several human diseases, such as atherosclerosis, inflammation, cancer, and, importantly, in neurological disorders, being relevant in the pathogenesis of retinopathies and glaucoma [31]. Thus, a better comprehension of Nb functional and biochemical properties could have important implications in understanding the molecular basis of these diseases and to offer novel therapeutic target(s).

## 2. Results

Mixing the Nb(II) solutions with NO_2_^−^ solutions (at pH 7.3 and 20.0 °C) induces a shift of the optical absorption maximum of the Soret band from 430 nm (i.e., *Mt*-Nb(II), *At*-Nb(II), *Dr*-Nb(II), and *Hs*-Nb(II)) to 414 nm (*Mt*-Nb(II)-NO), 408 nm (*At*-Nb(II)-NO), 401 nm (*Dr*-Nb(II)-NO), and 407 nm (*Hs*-Nb(II)-NO). Moreover, the absorbance spectrum of *Dr*-Nb(II)-NO showed a shoulder at 417 nm. The absorbance spectra of *Mt*-Nb(II)-NO, *At*-Nb(II)-NO, *Dr*-Nb(II)-NO, and *Hs*-Nb(II)-NO, obtained by NO_2_^−^ reduction, overlapped with those achieved by flowing gaseous NO in *Mt*-Nb(II), *At*-Nb(II), *Dr*-Nb(II), and *Hs*-Nb(II) solutions. Interestingly, the absorbance spectra of *Mt*-Nb(II)-NO, *At*-Nb(II)-NO, *Dr*-Nb(II)-NO, and *Hs*-Nb(II)-NO reflected the cleavage or the severe weakening of the proximal His-Fe(II) bond as evidenced from EPR spectroscopy [21,23,24].

Over the whole pH range explored (i.e., between pH 5.8 and 7.6), the kinetics of NO_2_^−^-based nitrosylation of *Mt*-Nb(II), *At*-Nb(II), *Dr*-Nb(II), and *Hs*-Nb(II) was fitted to a single-exponential decay according to Equation (3) (Figure 1, Figure 2, Figure 3 and Figure 4, panels A). According to the literature [27,28,29,32,33,34,35,36,37,38,39,40,41,42,43,44,45,46,47,48], this indicates that the Nb(III) intermediate species (see Figure 1), which is rapidly converted to Nb(II)-NO by reacting with dithionite and NO, does not accumulate in the course of the NO_2_^−^-based nitrosylation of *Mt*-Nb(II), *At*-Nb(II), *Dr*-Nb(II), and *Hs*-Nb(II).

Under all experimental conditions, the values of k_obs_ for the NO_2_^−^-based nitrosylation of Nb(II) increased linearly with NO_2_^−^ concentration (Figure 1, Figure 2, Figure 3 and Figure 4, panel B). The analysis of the data according to Equation (4) allowed the determination of *k*_on_ values (corresponding to the slope of the linear plots) for NO_2_^−^ (Table 1). The *y*-intercept of the linear plots was close to zero (Figure 1, Figure 2, Figure 3 and Figure 4, panel B), indicating that the NO_2_^−^-based nitrosylation of *Mt*-Nb(II), *At*-Nb(II), *Dr*-Nb(II), and *Hs*-Nb(II) can be considered as an essentially irreversible process, as already reported [27,28,29,30,32,36,40,41,42,44,45,46,47]. Moreover, the values of *k*_on_ for the NO_2_^−^-based nitrosylation of Nb(II) increased linearly with decreasing pH (Figure 1, Figure 2, Figure 3 and Figure 4, panel C) with slopes ranging between −0.96 ± 0.10 and −1.10 ± 0.10. This indicates the involvement of one proton in the NO_2_^−^-based conversion of Nb(II) to Nb(II)-NO (see Figure 1), as already reported for heme proteins and heme model compounds [27,28,29,30,32,36,40,41,42,44,45,46,47].

## 3. Discussion

In order to have an overall view of the nitrite reductase activity of heme proteins and of their structural determinants, a list of heme proteins has been reported in Table 1. As a starting point, the second-order rate constant of the nitrite reductase activity of the heme proteins was compared to that of CO binding, which is usually considered as a probe of the energetic barriers (both on the distal and proximal side) for the reactivity of the heme–Fe(II) atom. Moreover, the coordination of the metal center in the ferrous form, which should be of some help in finding out the main determinants of the reactivity, has been reported. For the sake of consistency, only the values of the bimolecular rate constants for CO binding obtained by the rapid-mixing technique, which only can be employed for measurement of the nitrite reductase activity, have been reported. Accordingly, the values of the bimolecular rate constant for CO binding reported in Table 1, which were only obtained by flash and laser photolysis, such as for ferrous six-coordinated plant Hbs (i.e., *Synechocystis* Hb (*S*-Hb(II)), rice nonsymbiotic Hb(II) class 1, and *Arabidopsis thaliana* Hb (*At*-Hb(II) class 1 and class 2), have been calculated according to Equation (1) (see also footnotes to Table 1):(1)$$k_{\mathrm{on}\left(\mathrm{CO}\right)}=\frac{k_{\mathrm{bind}\;}\times k_{\mathrm{in}\;}}{k_{\mathrm{in}\;}\times k_{\mathrm{out}\;}}\times \frac{k_{\mathrm{diss}\;}}{k_{\mathrm{diss}\;}+k_{\mathrm{ass}\;}}$$
where *k*_bind_ is the intrinsic rate of CO binding (as observed by geminate recombination), *k*_in_ and *k*_out_ are the rates of ligand entry and exit, respectively, from the heme pocket, *k*_diss_ is the dissociation rate of the endogenous six-coordinating ligand, and *k*_ass_ is its association rate; therefore, *k*_diss_ and *k*_ass_ account for the partial six-coordination of the species [49].

Figure 5 shows the correlation between the nitrite reductase activity and the CO-binding properties of heme proteins reported in Table 1. In the case of six-coordinated heme proteins, no apparent correlation was observed (Figure 5, panel A), even though rice nonsymbiotic Hb class 1 displayed the fastest rate constants of CO-binding and nitrite reductase activity (Table 1); this suggests that in rice nonsymbiotic Hb class 1, the axial six-coordinating bond with the endogenous ligand represents a low free energy barrier for both exogenous ligands (i.e., CO and NO_2_^−^). This occurrence might also be invoked for the relatively fast rate constants observed for other plant Hbs (i.e., *At*-Hb(II) class 1 and class 2 and *S*-Hb(II)), as compared to ferrous human neuroglobin (*Hs*-Ngb(II)) (Figure 5, panel A, and Table 1). On the other hand, in the case of ferrous human cytoglobin (*Hs*-Cygb(II)), which in the S-S monomeric form shows a nitrite reductase activity about six-fold faster than *At*-Hb(II) class 2 (in spite of an almost 200-fold slower rate constant for CO binding, see Figure 5, panel A, and Table 1), a different functional modulatory behavior must be taken into account. Thus, in *Hs*-Cygb(II), the disulfide bond between CysB2 and CysE9 plays a dramatic role in modulating the nitrite reductase activity, and the reduction of the CysB2-CysE9 bond brought about a 50-fold decrease of the nitrite reductase activity, an effect much more marked than for CO binding, where only a 5-fold reduction was observed (Figure 5, panel A, and Table 1). As a whole, in six-coordinated heme proteins, it looks like the His-Fe(II) axial sixth ligand regulates the barrier for CO binding, whereas in the case of nitrite reductase activity this is not the main determinant (as rice nonsymbiotic Hb(II) class 1 shows the fastest rate constant among all heme proteins investigated; see Table 1).

Conversely, in the case of five-coordinated heme proteins a linear correlation was observed between CO-binding rate constants and nitrite reductase activity (Figure 5, panel B), suggesting that several structural features allow the discrimination between CO and NO_2_^−^. As shown in Figure 5 (panel B), at least four classes of heme proteins have been identified. They may differ for the discrimination between the two ligands, as represented by the displacement along the *y*-axis of the straight lines in Figure 5 (panel B). Thus, five-coordinated heme proteins have been classified according to Equation (2) as follows:(2)$$r=\frac{k_{\mathrm{on}\left(\mathrm{CO}\right)}}{k_{\mathrm{on}\left({\mathrm{NO}}_{2}^{-}\right)}}$$

Class I (straight red line in Figure 5, panel B) groups heme proteins, which strongly discriminate between the two ligands, being characterized by *r* ≥ 2.5 × 10^6^ [i.e., the fast-reacting form of ferrous *Campylobacter jejuni* truncated HbP (*Cj*-trHbP(II)), human serum heme-albumin (*Hs*-heme(II)-albumin), *Equus ferus caballus* cytochrome *c* complexed with cardiolipin (*Efc*-Cytc(II)-CL), *Equus ferus caballus* microperoxidase 11 (*Efc*-MP11(II)), the fast-reacting form of *Methanosarcina acetivorans* protoglobin (*Ma*-Pgb(II)), and tetrameric human Hb(II) (*Hs*-Hb(II)) in the R quaternary state].

Class II (black straight line in Figure 5, panel B) groups heme proteins, which showed an intermediate discrimination power and *r* ≈ 5.6 × 10^5^ [i.e., tetrameric *Hs*-Hb(II) in the T-state, the slow-reacting form of *Ma*-Pgb(II), *Equus ferus caballus* carboxy-methylated cytochrome *c*(II) (carboxymethylated *Efc*-Cytc(II)), the dimeric human haptoglobin2-2:hemoglobin(II) complex (*Hs*-Hp2-2:Hb(II)), the slow-reacting form of *Cj*-trHbP(II), and the fast-reacting form of *Mycobacterium tuberculosis* truncated HbO(II) (*Mt*-trHbO(II))]. 

Class III (green straight line in Figure 5, panel B) groups heme proteins, which have less efficient discrimination between CO and NO_2_^−^, and are characterized by *r* ≈ 2 × 10^5^ [i.e., *Equus ferus caballus* Mb(II) (*Efc*-Mb(II)), *Physeter catodon* Mb(II) (*Pc*-Mb(II)), the dimeric human haptoglobin1-1:hemoglobin(II) complex (*Hs*-Hp1-1:Hb(II)), and *Mycobacterium tuberculosis* truncated HbN(II) (*Mt*-trHbN(II))].

Class IV (blue straight line in Figure 5, panel B) groups the slow-reacting form of *Mt*-trHbO(II) and all ferrous Nbs, which all showed a relatively poor discrimination power (*r* ≤ 2.5 × 10^4^).

The correlation emerging from the data in Figure 5 (panel B) indicated that within each class of heme proteins, a variation in CO-binding rate constant was accompanied by a similar behavior of the nitrite reductase activity, even though heme proteins belonging to different classes have a different way of discriminating between the two ligands. Thus, closely similar nitrite reductase activity between heme-proteins drastically differing for their CO binding behavior has been observed. A dramatic example is represented by tetrameric *Hs*-Hb(II) in the R-state, *Pc*-Mb(II) and *Hs*-Nb(II), which all displayed values of *k*_on(NO2−)_ ~ 6.0 M^−1^ s^−1^ while values of the CO-binding rate constant greatly differed, spanning between 1.0 × 10^7^ M^−1^ s^−1^ and 1.0 × 10^5^ M^−1^ s^−1^ (Figure 5, panel B, and Table 1). Although multiple conformations of both the distal and the proximal side of the heme pocket affect the CO-binding rate constant [79,80], a likely structural explanation has been attributed to the activation free energy for the ligand-induced movement of the Fe(II) atom into the heme plane, which is fairly low for tetrameric *Hs*-Hb(II) in the R-state, while it seems very high for *Hs*-Nb(II) [23,79]. As a matter of fact, it has been convincingly shown that a major contribution to the reactivity of CO with hemoproteins is represented by the energy required for the movement of the heme’s Fe atom from its unliganded position (about 0.5 Ǻ out of the heme plane on the proximal side) to the heme co-planar position in the CO-liganded form [2,79,81]; the major contribution stems from the steric repulsion between the imidazole of the proximal histidine and the heme pyrroles, which depends on the relative position and differs among various hemoproteins [2]. Obviously, the conformation of the distal portion of the heme pocket is also important to account for the different CO-binding behaviors shown by the various hemoproteins [80], but it is the variation of the activation free energy for the ligand-linked movement of the Fe–His bond which accounts for the modulation of the CO-binding kinetics of a specific protein by environmental conditions, such as pH [81].

On the basis of these considerations, class I may be representative of heme proteins with a very low activation free energy on both the proximal and the distal side of the heme pocket. It is illustrative that they also display a fairly fast nitrite reductase activity (Figure 5, panel B, and Table 1); the slight variation within this class is due to small variations of the heme distal side conformation, affecting the kinetics for both ligands. On the other hand, class II and III appear to be heme proteins displaying a high proximal barrier for the reaction with CO, thus slowing down the carbonylation rate constant; however, this barrier does not affect dramatically the nitrite reductase activity. Therefore, differences in nitrite reductase activity within heme proteins belonging to class II and III is likely attributable to distal barriers. Lastly, class IV includes heme proteins (such as the slow-reacting form of the *Mt*-trHbO(II) and all ferrous Nbs), which have a very high free energy proximal barrier, which dramatically slows down the CO-binding rate constant. Consequently, differences in the nitrite reductase activity are probably due to a much higher distal barrier in *Mt*-trHbO(II) than in all ferrous Nbs, which display a very open heme pocket and a fairly fast nitrite reductase activity (Figure 5, panel B, and Table 1). As a whole, different classes, reported in Figure 5 (panel B), reflect various free energy proximal barriers for CO-binding whereas different positions along the same line would refer to variations in the free energy distal barriers.

Even among ferrous six-coordinated heme proteins [i.e., *Hs*-Cygb(II) and *Mus musculus* Ngb(II) (*Mm*-Ngb(II)], the discriminatory power varied dramatically (*r* < 4.5 × 10^5^; Figure 5, panel A, and Table 1). The low value of *r* depends on the low CO-binding rate constant, reflecting the occupancy of the sixth axial heme coordination by the heme distal histidyl residue and the strength of the axial Fe(II)–His distal bond. This is likely responsible for the variation in the nitrite reductase activity among the various six-coordinated heme proteins (Figure 5, panel A, and Table 1). In particular, the longer the Fe-His proximal and distal bonds are in six-coordinated *S*-Hb(II), rice nonsymbiotic Hb(II) class 1, and *At*-Hb(II) class 1, as compared to bis-histidyl cytochromes, the lower the bond strength is. This is likely a factor enabling the heme distal His dissociation and the subsequent binding of exogenous ligands in six-coordinated Hbs [82]. Moreover, six-coordinated *S*-Hb(II), rice nonsymbiotic Hb(II) class 1, and *At*-Hb(II) class 1 displayed larger tilt angles for both the proximal and distal His residues compared with cytochrome b5. This decreases the strength of the heme–Fe-His bond contributing to fast ligand binding of these six-coordinated globins, likely playing a role of the utmost importance in characterizing their high nitrite reductase activity (Figure 5, panel A, and Table 1) [82,83,84,85].

## 4. Materials and Methods

*Mt*-Nb(III), *At*-Nb(III), *Dr*-Nb(III), and *Hs*-Nb(III) were prepared as previously reported [16,17,19,21,25]. The concentration of *Mt*-Nb(III), *At*-Nb(III), *Dr*-Nb(III), and *Hs*-Nb(III) was determined spectrophotometrically at λ = 407 nm, the values of ε being 100 mM^−1^ cm^−1^, 160 mM^−1^ cm^−1^, 157 mM^−1^ cm^−1^, and 147 mM^−1^ cm^−1^, respectively [19,21,23,24]. *Mt*-Nb(II), *At*-Nb(II), *Dr*-Nb(II), and *Hs*-Nb(II) solutions were obtained by adding dithionite solution (final concentration, 3.0 × 10^−3^ M) to *Mt*-Nb(III), *At*-Nb(III), *Dr*-Nb(III), and *Hs*-Nb(III) (final concentration ranging between 2.2 × 10^−6^ M and 3.1 × 10^−6^ M) under anaerobic conditions. Gaseous NO (Linde Caracciolossigeno S.r.l., Roma, Italy) was purified by flowing through a column packed with NaOH pellets and then by passage through a 5.0 M NaOH trapping solution to remove acidic nitrogen oxides; the NO pressure was 760.0 mmHg [86]. The stock NO solution was prepared anaerobically by keeping the degassed 1.0 × 10^−2^ M 1,3-bis(tris(hydroxymethyl)methylamino)propane buffer solution (pH 7.0) in a closed vessel under NO at *p* = 760.0 mmHg and 20.0 °C [1]. All the other chemicals were purchased from Merck KGaA (Darmstadt, Germany) and Sigma-Aldrich (St. Louis, MO, USA). All chemicals were of analytical grade and were used without further purification unless stated otherwise.

The absorbance spectra of *Mt*-Nb(II)-NO, *At*-Nb(II)-NO, *Dr*-Nb(II)-NO, and *Hs*-Nb(II)-NO were obtained either by adding nitrite (final concentration, 2.0 × 10^−2^ M) to *Mt*-Nb(II), *At*-Nb(II), *Dr*-Nb(II), and *Hs*-Nb(II) or by flowing gaseous NO (final concentration 1.0 × 10^−4^ M) into the Nb(II) solutions. The final concentration of *Mt*-Nb(II), *At*-Nb(II), *Dr*-Nb(II), and *Hs*-Nb(II) ranged between 2.2 × 10^−6^ M and 3.1 × 10^−6^ M.

The kinetics of NO_2_^−^ reduction from *Mt*-Nb(II), *At*-Nb(II), *Dr*-Nb(II), and *Hs*-Nb(II) (i.e., of Nb(II)-NO formation) were analyzed in the framework of the minimum reaction mechanism depicted in Figure 1 [27,29,32,33,34,35,36,37,38,39,40,41,42,43,44,45,46,47,48,53].

The values of the apparent pseudo-first-order rate constant (i.e., *k*_obs_) for NO_2_^−^ reduction from *Mt*-Nb(II), *At*-Nb(II), *Dr*-Nb(II), and *Hs*-Nb(II) were determined by rapid-mixing the heme protein solutions (final concentration ranging between 2.2 × 10^−6^ M and 3.1 × 10^−6^ M) with the NO_2_^−^ solution (final concentration ranging between 2.0 × 10^−3^ M and 2.0 × 10^−2^ M) in the presence of sodium dithionite (final concentration, 3.0 × 10^−3^ M). A sodium dithionite concentration lower than 1.0 × 10^−2^ M neither reduces NO_2_^−^ to NO [37] nor reacts with NO [87]. No gaseous phase was present. The NO_2_^−^ reduction by *Mt*-Nb(II), *At*-Nb(II), *Dr*-Nb(II), and *Hs*-Nb(II) was monitored spectrophotometrically between 380 and 450 nm, with a wavelength interval of 5 nm. The values of *k*_obs_ were obtained according to Equation (3) [27,28,29,30,32,36,40,41,42,44,45,46,47]:(3)$${\left[\mathrm{Nb}\left(\mathrm{II}\right)\right]}_{t}={\left[\mathrm{Nb}\left(\mathrm{II}\right)\right]}_{i}\times e^{-k\mathrm{obs}\;\times t}$$
where Nb(II) is either *Mt*-Nb(II), *At*-Nb(II), *Dr*-Nb(II), or *Hs*-Nb(II).

The values of the apparent second-order rate constant for NO_2_^−^ reduction by *Mt*-Nb(II), *At*-Nb(II), *Dr*-Nb(II), and *Hs*-Nb(II) (i.e., *k*_on_) were determined from the linear dependence of *k*_obs_ on the NO_2_^−^ concentration (i.e., [NO_2_^−^]), according to Equation (4) [27,28,29,30,32,36,40,41,42,44,45,46,47]:(4)$$k_{\mathrm{obs}}=k_{on}\times [{\mathrm{NO}}_{2}^{-}]$$

The time courses of NO_2_^−^ reduction from *Mt*-Nb(II), *At*-Nb(II), *Dr*-Nb(II), and *Hs*-Nb(II) were obtained with a SFM-20/MOS-200 rapid-mixing stopped-flow apparatus (BioLogic Science Instruments, Claix, France).

The kinetic parameters were obtained between pH 5.8 and 7.6 (5.0 × 10^−2^ M of 2-(*N*-morpholino)-ethanesulfonic acid between pH 5.8 and 6.6, and 5.0 × 10^−2^ M 1,3-bis(tris(hydroxymethyl)-methylamino)propane between pH 6.3 and 7.6), at 20.0 °C. The different buffers did not affect the values of the kinetic parameters obtained at overlapping pH values.

The kinetic and thermodynamic data were analyzed with the Prism 5.03 program (GraphPad Software, Inc., La Jolla, CA, USA). The results are given as mean values of at least four experiments plus or minus the corresponding standard deviation.

## 5. Conclusions

All Nbs show a fairly high nitrite reductase activity, a property which strengthens the hypothesis that they are mostly involved in the NO metabolism [18,20]. This enzymatic activity of heme proteins is one of the most efficient ways for the production of NO starting from the reduction of NO_2_^−^, a pivotal process for the regulation of blood vessel muscular tone and the regulation of the blood flow. Of note, in the retina, NO levels are crucial to maintain normal visual functions, being relevant for photoreceptor light transduction and the control of retinal blood flow, opening a perspective on a major role of Nbs in retinal disorders [88,89]. Moreover, a link between NO and Nb-based signaling and chemistry has been envisaged in *M*. *tuberculosis*, *A*. *thaliana*, *D*. *rerio* and *H. sapiens*. Specifically, the survival of *M*. *tuberculosis* in the host implies the presence of effective detoxification systems, including Nbs, to inactivate RNS and ROS produced by the immune response [19,25,26]. In *A*. *thaliana*, Nb has been hypothesized to transport and release NO at the infection site; moreover, NO may reduce the superoxide radical with the generating peroxynitrite that increases pathogen burden [16,19]. Interestingly, *Dr*-Nb may play a relevant physiological role in peroxynitrite scavenging from poorly oxygenated tissues, such as the retina in fish where blood circulation is critical for adaptation to diving conditions [21,22,26]. It is worth remarking that *Danio rerio* Nb shows the fastest nitrite reductase activity (Table 1), outlining the fact that, in fishes, the O_2_ supply to poorly oxygenated tissues, such the retina, occurs by means of a fine regulation of the eye circulation through the *rete mirabilis*, with NO playing a major role in regulating the blood flow and thus the oxygenation of retinal layers [90]. Finally, *Hs*-Nb represents the *C*-terminal domain of the nuclear protein named THAP4, which displays a *N*-terminal modified zinc finger domain that binds DNA. Since *Hs*-Nb(III) binds NO without recognizing CO and O_2_, the Nb domain may be relevant for a NO-linked selective modulation of gene transcription [19,21,24].

Here, we report a comparison of a large number of heme proteins with drastically different conformations of the heme cavity, which casts light on the structure–function relationships, which modulate the nitrite reductase activity. In particular, we identified the accessibility of the heme distal pocket as an important factor since various heme proteins with different proximal constraints showed similar nitrite reductase activity. On the other hand, heme-proteins displaying different distal structural arrangements, but similar proximal constraints, show a remarkable effect on this enzymatic activity. However, other factors, such as the redox potential, cannot be discarded since the NO_2_^−^ reduction to NO requires transient heme oxidation (see Figure 1); unfortunately, for many of the investigated heme proteins this information is not yet available and a thorough comparison is presently not possible.

## Data Availability

Not applicable.

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
