# Peer review of "Nitrite Reductase Activity of Ferrous Nitrobindins: A Comparative Study"

_ijms, 2023, doi:10.3390/ijms24076553_

Round 1
Reviewer 1 Report
This is a generally well written paper in which the authors present new data on the kinetics of CO binding and the nitrite reductase activity of four nitrobindins. The experiments have been carefully performed and the results sensibly analysed and presented. In addition, the authors have drawn together a very large body of data on these kinetics from an extensive range of heme proteins. These data have allowed the authors to compare konCO with konNO2- and, for penta coordinate heme proteins, to group these into four classes. This is an interesting synthesis and of interest to readers. The discussion is useful and the authors suggest structural and dynamic reasons underpinning the behaviour of these classes.
The authors should consider the following suggestions for improvement of the manuscript prior to publication.
The discussion that deals with the underlying reasons that allow the proteins to be grouped according to ϒ is rather densely written. It would be of advantage to the reader if this section were supplemented with a schematic diagram/model illustrating the structural motifs and dynamics that constitute the proposed energy barriers discussed and which are responsible for the protein grouping seen in Fig 5B. If space is limited, room may be found by condensing Figs 1-4 which, apart from numerical values, are all essentially the same.
Minor point and typographical suggestions.
Where does Eq1 come from? I don’t think it in Smagghe 2006.
Line 269 “where” not “wherefore”.
Line 271 delete initial “that”.
Line 303 Not “envisaged” perhaps “identified”?
Lines 310, 317, 324. Not “referable” but “refers”.
Line 349 Not “is referable to” but perhaps “reflects”.
Line 351 Not “refer to” but “be”.
Line 361 Delete “would”.
Line 466 Better “opening a perspective on the”
Author Response
Reviewer 1.1. The discussion that deals with the underlying reasons that allow the proteins to be grouped according to ϒ is rather densely written. It would be of advantage to the reader if this section were supplemented with a schematic diagram/model illustrating the structural motifs and dynamics that constitute the proposed energy barriers discussed and which are responsible for the protein grouping seen in Fig 5B.
Author 1.1. We have amplified the part concerning the logics behind the choice of the four classes of hemoproteins, based on the different CO binding behavior. This is marked in yellow in the Discussion of the revised version (lines 320-331).
Reviewer 1.2. If space is limited, room may be found by condensing Figs 1-4 which, apart from numerical values, are all essentially the same.
Author 1.2. Since the space is not limited, Figs 1-4 have not been grouped, representing the experimental data referring to the four nitrobindins investigated. Thus, although they are similar (but not identical), we deem important to show separately the kinetic properties of each nitrobindin.
Reviewer 1.3. Where does Eq1 come from? I don’t think it in Smagghe 2006.
Author 1.3. Our eq. (1) has been formulated by us with parameters extractable from laser photolysis data, but it is conceptually analogous to eq. (1) from Smagghe et al., 2006; thus, we employed our eq.(1) also for data from other authors. The first term is the rate constant from stopped-flow and the second term accounts for the hexa-coordination of the unliganded form. In any case, we think that description of the parameters, following the equation itself, is sufficiently self-explanatory.
Reviewer 1.4. Line 269 “where” not “wherefore”.
Author 1.4. According to the Reviewer request in the line 269 “wherefore” has been replaced with “where”.
Reviewer 1.5. Line 271 delete initial “that”.
Author 1.5. According to the Reviewer request in the line 271 the initial “that” has been deleted.
Reviewer 1.6. Line 303 Not “envisaged” perhaps “identified”?
Author 1.6. According to the Reviewer request in the line 303 “envisaged” has been replaced with “identified”.
Reviewer 1.7. Lines 310, 317, 324, 349. Not “referable” but “refers”.
Author 1.7. According to the Reviewer request in the lines 310, 317, 324, 349 “referable” has been replaced with “refers”.
Reviewer 1.8. Line 351 Not “refer to” but “be”.
Author 1.8. According to the Reviewer request in the line 351 “refer to” has been replaced with “be”.
Reviewer 1.9. Line 361 Delete “would”.
Author 1.9. According to the Reviewer request in the line 361 “would” has been deleted
Reviewer 1.10. Line 466 Better “opening a perspective on the”.
Author 1.10. According to the Reviewer request in the line 465 “opening to perspective of” has been replaced with “opening a perspective on the”.
Reviewer 2 Report
Authors present a comparative study about kinetics of nitrite reductases. They performed well manage studies by using stopped-flow on 4 enzymes and also compared their results with former studies described in the literature. The experiments were informative and the authors discussed their results and compared them in Figure 5 that gives a lot of information about the discrimination between CO and NO by enzymes. The study bring novel data and enriched discussion about biological relevance.
Author Response
The text has been carefully revised in agreement with Reviewer’s suggestions to make it flowing and clear. In the “marked manuscript”, all changes are highlighted in yellow.
We are grateful to the Reviewers since their comments helped us to improve some important concepts in our manuscript as well as the presentation of data.
Best regards
Massimo Coletta and Paolo Ascenzi
Reviewer 3 Report
The manuscript describes the nitrite reductase activity of ferrous nitrobindins (Nbs II), which are present in bacteria, plants, fish and humans. They catalyse the reduction of NO2⁻ to NO with the concomitant oxidation of Nb(II) to Nb(III). The assays contained dithionite, which immediately reduced Nb(III) + NO to Nb(II)-NO. The difference of the absorption of Nb(II) and N(II)-NO at an appropriate wavelength was measured. The authors compare their results with the nitrite reductase activity of haemoglobins and emphasize the importance of the formed NO triggering various physiological processes. Besides the rate constants of four different Nbs, the manuscript contains only data from other papers.
The manuscript is difficult to assess, because on several places a huge amount of citations up to 4 lines, mainly from the authors’ own laboratory, disturb the reading flow. The table contains besides kon the unnecessary logkon, while necessary data such as pH and temperature are found in footnotes. All four figures can be reduced to one, because they are almost identical. Specific notes:
Line 16: peroxynitrite isomerization to NO3‒ and NO3- !
Line 22: change 1.4 × 101 M‒1 s‒1 to 14 M‒1 s‒1
Line 143 (Fig. 1, also Figs 2 and 4): The value of the slope of the continuous line is: −0.96 ± 0.1; change to −0.96 ± 0.10 or -1.0 ± 0.1.
Author Response
Reviewer 3.1.
The manuscript is difficult to assess, because on several places a huge amount of citations up to 4 lines, mainly from the authors’ own laboratory, disturb the reading flow.
Author 3.1. According to the paper style, extended citations have been replaced with reference numbering.
Reviewer 3.2.
The table contains besides kon the unnecessary logkon, while necessary data such as pH and temperature are found in footnotes.
Author 3.2. Values of logkon have been removed from Table 1. Values of pH and temperature cannot be reported in Table 1 since they often refer to different parameters on the same line.
Reviewer 3.3. All four figures can be reduced to one, because they are almost identical.
Author 3.3. Figures 1-4 have not been grouped since they represent the experimental data referring to the four nitrobindins investigated. Thus, although they are similar (but not identical), we deem important to show separately the kinetic properties of each nitrobindin.
Reviewer 3.4. Line 16: peroxynitrite isomerization to NO3‒ and NO3‒ !
Author 3.4. Lines 14-16: According to the Reviewer request, the sentence at line 16 has been corrected as follows: “They inactivate reactive nitrogen species by sequestering NO, converting NO to HNO2, and pro-moting peroxynitrite isomerization to NO3‒”.
Reviewer 3.5. Line 22: change 1.4 × 101 M‒1 s‒1 to 14 M‒1 s‒1
Author 3.5. Line 22: For homogeneity within the whole text, the value 1.4 × 101 M‒1 s‒1 has not been replaced with 14 M‒1 s‒1.
Reviewer 3.6. Line 143 (Fig. 1, also Figs 2 and 4): The value of the slope of the continuous line is: −0.96 ± 0.1; change to −0.96 ± 0.10 or -1.0 ± 0.1.
Author 3.6. According to the Reviewer request, the values of the slope of the continuous lines have been changed from −0.96 ± 0.1, −1.03 ± 0.1, –1.1 ± 0.1, and −0.99 ± 0.1 to −0.96 ± 0.10, −1.03 ± 0.10, –1.1 ± 0.10, and −0.99 ± 0.10, respectively, within the whole text.